# Redox-neutral photocatalytic hydrodealkenylation of aryl olefins

Ke Liao[1], Chunming Gui[1], Ziming Cao[2], Yong Huang [3] ✉ & Jiean Chen [1] ✉

Carbon-carbon bond cleavage is a transformative strategy in chemical synthesis, particularly for modifying complex molecules. While the cleavage of $C(sp^2)=C(sp^2)$ π-bonds is relatively straightforward, the selective cleavage of unpolarized $C(sp^2)–C(sp^3)$ σ-bonds remains a significant challenge. In this study, we present a redox-neutral approach for hydrodealkenylation, enabling the selective cleavage of $C(sp^2)–C(sp^3)$ σ-bonds in aryl olefins. This reaction proceeds via a cascade of aryl radical cation-mediated open-shell steps under photoredox conditions, incorporating an alkene migration step that exhibits high selectivity and synthetic versatility. This protocol facilitates the selective removal of a vinyl group from arylalkene substrates, yielding isolable fragments. Moreover, this method extends beyond single-bond cleavage by enabling a domino reaction sequence capable of cleaving multiple inert carbon-carbon σ-bonds and allowing programmable chain homologation. This work advances the field of σ-bond cleavage and functionalization, offering a versatile tool for the molecular editing of hydrocarbons with significant implications for synthetic chemistry and the development of novel chemical transformations.

Alkenes are among the most prevalent biomass and industrial feedstock compounds[1–3]. Consequently, extensive research has focused on the chemical transformation of alkenes into a diverse array of functional groups[4–8]. The selective cleavage and subsequent functionalization of carbon-carbon bonds in alkenes have become crucial in organic synthesis. Significant advancements have been made in cleaving olefinic bonds, particularly $C(sp^2)=C(sp^2)$ bonds, through oxidation reactions (Fig. 1A)[9–13]. However, the direct cleavage of C−C σ bonds remains a formidable challenge due to the ubiquitous nature of similar carbon-carbon σ bonds in alkenes, which include multiple $C(sp^3)–C(sp^3)$ and $C(sp^2)–C(sp^3)$ bonds within a single molecule. While the cleavage of allylic $C(sp^3)–C(sp^3)$ bonds has been moderately successful, primarily facilitated by transition-metal-catalyzed retroallylation and deallylation reactions (Fig. 1B)[14,15], the cleavage of $C(sp^2)–C(sp^3)$ bonds is primarily limited to polar variants, such as acyl-$C(sp^3)$ bonds[16–20]. Methods for cleaving nonpolar $C(sp^2)–C(sp^3)$ bonds have significantly lagged, highlighting an area in urgent need of advancement[21–24].

The Kwon group has significantly contributed to this field by developing a $Fe^{(II)}/O_3$ system for generating oxyradicals under redox conditions, leading to dealkenylation products via β-scission[25]. Subsequently, they reported versatile transformations leveraging in-situ generated alkyl radicals[26–30]. Stepwise protocols under low temperatures are commonly required to enhance reaction efficiency. This oxidative strategy typically involves cleavage at the multi-substituted site in scenarios where the olefin is multi-substituted. A redox-neutral approach is needed as a complementary strategy to achieve selective cleavage at the less-substituted site (Fig. 1C). Our group has focused on selectively cleaving unstrained carbon-carbon bonds, establishing robust protocols for photocatalytically cleaving unactivated $C(sp^2)=C(sp^2)$ and $C(sp^3)–C(sp^3)$ bonds via single-electron oxidation strategies[31,32]. However, cleaving $C(sp^2)–C(sp^3)$ bonds remains particularly challenging due to their greater strength compared to typical $C(sp^3)–C(sp^3)$ bonds. The inclusion of an aryl group further complicates selectivity between $C(aryl)–C(sp^3)$ and $C(vinyl)–C(sp^3)$ bonds,

[1]Pingshan Translational Medicine Center, Shenzhen Bay Laboratory, Shenzhen, China. [2]College of Pharmacy, Shenzhen Technology University, Shenzhen, China. [3]Department of Chemistry, The Hong Kong University of Science and Technology, Clear Water Bay, Kowloon, Hong Kong SAR, China. ✉e-mail: yonghuang@ust.hk; chenja@szbl.ac.cn

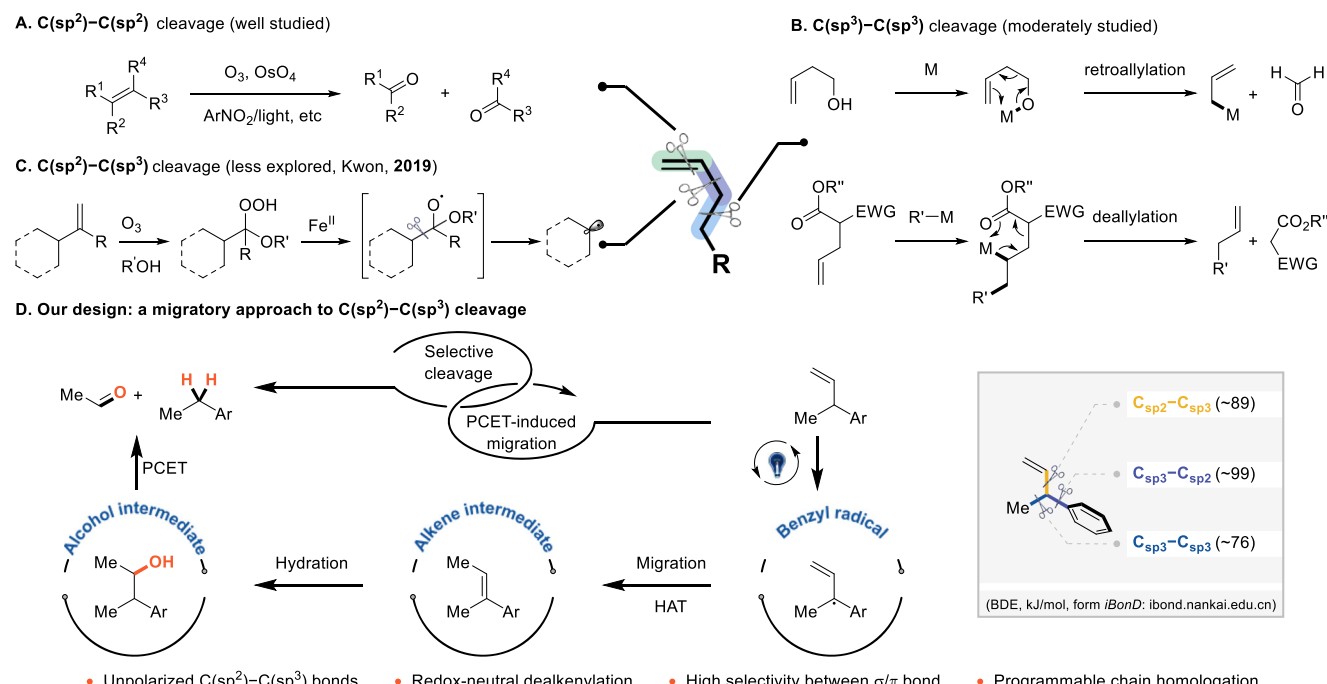

**Fig. 1 | Cleavage of unactivated carbon-carbon bonds. A** Oxidative cleavage of C(sp²)=C(sp²) bonds. **B** Transition-metal-catalyzed cleavage of allylic C(sp³)-C(sp³) bonds. **C** Oxidative cleavage of C(sp²)-C(sp³) bonds. **D** Redox-neutral cleavage of C(sp²)-C(sp³) bonds (this work). PCET photo-coupled electron transfer, HAT hydrogen-atom transfer.

especially given their similar bond dissociation energies (BDE). Hence, a PCET/HAT-induced olefin isomerization strategy is proposed to address the challenge brought by bond similarity.

In this work, we present a photoredox approach to the selective cleavage of C(sp²)−C(sp³) bonds in arylalkenes (Fig. 1D). The transformation proceeds through several fundamental steps: Light-mediated single-electron oxidation (SEO) generates the corresponding aryl radical cation intermediate. Proton-coupled electron transfer forms a benzylic radical. This radical is captured by a hydrogen atom transfer (HAT) reagent at the resonance carbon. Subsequent photoredox cleavage of the C(sp²)=C(sp²) bond achieves the desired bond cleavage. The selection of HAT reagents is important as they may directly affect the benzyl radical, arousing the challenge of inhibiting the migration. This strategy effectively deletes a vinyl group from the substrate, converting it into an aldehyde. The overall redox-neutral nature of this approach is particularly appealing, as it incorporates one molecule of water between the two fragments, yielding two valuable products. However, the key challenge lies in controlling the reaction pathway. The aryl radical cation is prone to various competing processes, such as proton-coupled electron transfer (PCET), carbocation-coupled electron transfer (CCCET), and nucleophilic addition[33–36]. Proper sequencing of each step is crucial, as any misalignment can lead to entirely different products.

## Results

### Reaction optimization

We initiated our investigation using 1-allyl-4-tert-butylbenzene (**1a**) as the substrate and a Fukuzumi/Nicewicz acridinium analog (PC-1) as the photoredox catalyst. This catalyst is renowned for its efficient oxidizing high-potential aryls under 450 nm LED light, producing the corresponding aryl radical cations[37–39]. Despite numerous attempts, the devinylation product **2** was detected only in trace amounts (Table 1, entry 1). However, a significant improvement was observed with the addition of 1/10 (v/v) water to the reaction mixture (entry 2), suggesting that direct C(sp²)−C(sp³) bond cleavage does not readily occur for aryl radical cations. Water was necessary to facilitate bond

cleavage via β-scission. Next, we evaluated the impact of the HAT reagent. Consistent with our expectations, the yield of product **2** was highly dependent on the HAT reagent selected, with pentafluorobenzenethiol (HAT-3) proving to be the most efficient option (entry 4). Additionally, we confirmed that HAT-3 could be used catalytically, albeit with a slightly reduced yield (entry 7). The reaction exhibited single selectivity between σ/π bonds as product **3** was not detected.

We rapidly assessed substituted alkenes and found the reaction remarkably effective across all olefin substitution patterns, including tetrasubstituted ones. Contrary to our initial speculation that multi-substituted alkenes might resist radical-mediated migration, introducing two terminal methyl groups did not hinder bond cleavage. We anticipated that the HAT reagent would face difficulty accessing the hindered terminal carbon in multi-substituted olefins, making olefin migration less favorable both kinetically and thermodynamically. Notably, substrate **1d** exhibited only a slight decrease in yield, underscoring the strong adaptability of the reaction to a variety of substituted alkenes. Control experiments confirmed that the HAT reagent, photocatalyst, and light are indispensable components for the reaction to proceed.

### Substrate scope

The scope of devinylation was investigated under the optimal conditions listed in Fig. 2. Various aryl substituents exhibited good tolerance, particularly those with multiple benzylic carbons, which are notable for generating multiple benzylic radical intermediates through PCET, leading to selectivity challenges[40]. The devinylation products were obtained in good-to-high yields (products **2a-2v**). Redox-active functional groups, known for their versatile open-shell reactivities–such as thiophene, dibenzofuran, phthalimide, and alcohol–showed good compatibility with acridinium photocatalysts.

The mild reaction conditions allowed the use of compounds containing pH-sensitive groups, including lactones, phenyl esters, ketals, and hemiacetals. Substrates featuring branched benzylic carbons yielded their respective secondary alkyl radicals, resulting in

## Table 1 | Optimization of reaction conditions[a]

| entry | substrate | HAT (eq.) | yield (%) |
|---|---|---|---|
| 1[b] | 1a | HAT-1 (1.0 eq.) | trace |
| 2 | 1a | HAT-1 (1.0 eq.) | 50 |
| 3 | 1a | HAT-2 (1.0 eq.) | 0 |
| 4 | 1a | HAT-3 (1.0 eq.) | 93 |
| 5 | 1a | HAT-4 (1.0 eq.) | 20 |
| 6 | 1a | HAT-3 (0.5 eq.) | 81 |
| 7 | 1a | HAT-3 (0.25 eq.) | 69 |
| 8 | 1b | HAT-3 (1.0 eq.) | 74 |
| 9 | 1c | HAT-3 (1.0 eq.) | 89 |
| 10 | 1d | HAT-3 (1.0 eq.) | 39 |
| 11 | 1a | — | 0 |
| 12[c] | 1a | HAT-3 (1.0 eq.) | 0 |

[a]A two-dram glass vial containing a mixture of PC (0.005 mmol, 2.5 mol%), substrate (0.2 mmol), HAT (0.2 mmol) in MeCN/water (1.0 mL/0.1 mL) was placed between two Kessil LED lights (440 nm, 40 W) and vigorously stirred for 15 h at 40 °C. Yields were determined by GC analysis using an external standard. [b] Water was omitted. [c] Light or PC was omitted. PC photocatalyst, HAT hydrogen-atom transfer.

benzene derivatives with higher-order alkyl substituents (products **2w-2ad**). These findings underscore the effectiveness of this transformation, especially considering that branched benzylic carbons are documented to undergo direct $C(sp^3)$–$C(sp^3)$ cleavage (CCCET). Furthermore, several complex molecules with intricate structures underwent smooth cleavage of the $C(sp^2)$–$C(sp^3)$ bonds, successfully generating hydrodevinylative analogs (products **2q-2s, 2ab-2ac**).

When the two aryl groups exhibit subtle differences in electron density, exclusive C–C bond cleavage occurs on the side of the electron-rich aryl group. This behavior is presumably due to the thermodynamic preference for alkene migration to conjugate with an electron-rich phenyl group. Notably, simply switching the methyl group from one side to the other resulted in a complete switch at the site of bond cleavage, from $C(sp^2)$–$C(sp^3)$ to $C(sp^2)$=$C(sp^2)$ (substrates **2ae** and **2af**). The introduction of electron-withdrawing groups, such as cyano (CN) or fluoro (F), on the aryl moieties elicited similar effects (substrates **2ag** and **2ah**).

In this study, we successfully isolated the cleaved fragment as the corresponding aldehyde in good-to-high yields. This result is noteworthy, as it contrasts with the classical oxidative cleavage of alkenes, which predominantly yields ketones—the additional carbon atom in the product results from olefin migration (Fig. 3).

We investigated the scope of the olefin terminus by altering the $R^1$ and $R^2$ groups. Linear aldehydes are produced when either $R^1$ or $R^2$ is a hydrogen atom. Conversely, when $R^1$ and $R^2$ are non-hydrogen groups, α-branched aldehydes are formed. The reaction demonstrated commendable accommodation of various functional groups. Ether, ester,

imide, and sulfamide groups—typically labile under hydrogen atom transfer (HAT) conditions—were well tolerated in this reaction (products **3b, 3 d, 3 l, 3r**). Moreover, the presence of an additional olefin did not interfere with the migration and bond cleavage processes, remaining intact after the reaction (product **3 s**). Complex structures also reacted smoothly under these conditions, yielding the desired products (product **3t**).

### Mechanistic studies

Subsequent experiments were conducted to elucidate the mechanism underlying the photoredox-induced dealkenylation reaction. A light-on-off experiment ruled out the involvement of radical chain mechanisms (Fig. 4A). The reaction rates increased with higher light intensity, establishing a positive correlation (Fig. 4B). The Stern-Volmer quenching experiments confirmed that the alkene was the primary quencher of the excited photocatalyst (Supplementary Fig. 8). A radical trapping experiment using benzalmalononitrile confirmed the generation of a benzyl radical following key carbon-carbon bond cleavage (Fig. 4C). Further evidence for the alkene-migratory mechanism was obtained by replacing water with $D_2O$, which yielded a dealkenylation product with a D/H ratio of 1.68/1.32 at the benzylic position, supporting the proposed mechanism involving alkene migration and two HAT events (Figs. 4D, d-**4a**).

To substantiate the role of alkene migration, a styrene analog **4a'** was treated under standard reaction conditions, resulting in efficient cleavage of the $C(sp^2)$=$C(sp^2)$ bond. Additionally, the radical cation-mediated hydration product **4a-OH** was suggested as an intermediate

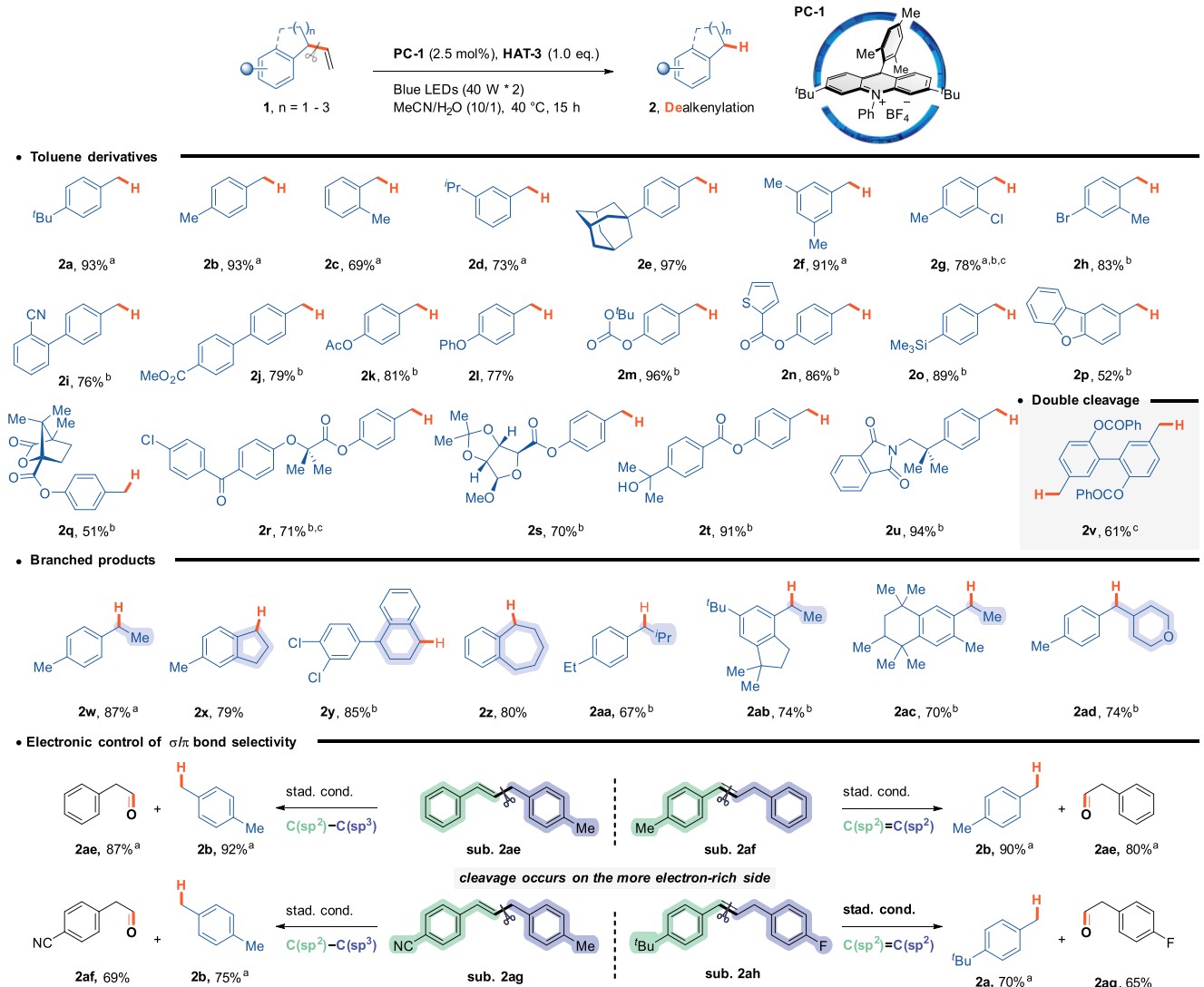

**Fig. 2 | The generality of the devinylation reaction.** A two-dram glass vial containing a mixture of PC-1 (0.005 mmol, 2.5 mol%), substrate (0.2 mmol), and HAT-3 (0.2 mmol) in MeCN/water (1.0 mL/0.1 mL) was placed between two Kessil LED lights (440 nm, 40 W) and vigorously stirred for 15 h at 40 °C. **a** Yields were determined by GC because of the low boiling points of these products. **b** Temperature: 50 °C. **c** Reaction time: 48 h. PC photocatalyst, HAT hydrogen-atom transfer.

in this transformation. When this alcohol was subjected to the reaction conditions, the desired bond cleavage product was formed quantitatively (Fig. 4E). Control experiments without water enabled the detection of the migratory intermediate via GCMS analysis (**4a'**). This observation aligns with kinetic studies confirming the formation of alcohol (**4a-OH**) derivatives originating from migratory styrene, supporting the proposed radical migration pathway (Supplementary Fig. 3, Table 3 and Fig. 7). These results collectively support the sequential formation of these crucial intermediates before carbon-carbon bond cleavage occurs.

To further validate the alkene migration step, we tested substrates that did not undergo such migration, particularly those lacking benzylic protons (substrate **4b**) and homoallylic arenes (substrate **4c**). In these cases, the starting materials remained largely unreacted (Fig. 4F). The experimental data gathered thus far strongly support a reaction cascade involving photocatalytic alkene migration (PCET followed by HAT), hydration of the conjugated alkene via SEO, and subsequent C(sp³)−C(sp³) bond cleavage via PCET, β-scission, and HAT (Supplementary Figs. 10 and 11).

To overcome the reactivity challenges associated with non-migratory arylalkenes, we explored introducing additional substituents into the olefin component. The oxidation potential of olefins is known to decrease with an increasing number of substituents. Tri- and tetra-substituted alkenes have been reported to undergo direct SEO via excited acridinium salts[41,42]. By incorporating extra methyl groups, the substituted alkenes were positioned to potentially compete with the arene moiety for SEO mediated by the excited photocatalyst. Gratifyingly, this strategic modification successfully restored the desired reactivity, facilitating C(sp²)−C(sp³) bond cleavage and affording dealkenylation products in good yields (Fig. 4G, products **4c-4h**). Notably, nonsubstituted and *para*-bromide-substituted alkenes can be cleaved (products **4 d, 4e**). Further investigation of more electron-deficient alkenes indicates that the reaction will be interrupted at the stage of hydration, which is consistent with our previous study: the SEO of aromatic rings is indispensable for PCET-induced C-C bond cleavage[39]. Notably, homoallyl arenes, which frequently possess multiple benzylic protons, are susceptible to PCET and can interfere with the intricate cascade of photoredox events. Despite this complexity, the sequential formation of key intermediates was meticulously preserved (products **4 g, 4 h**).

The migratory cleavage mechanism presents a distinct opportunity to achieve domino cleavage. Phenyl radical cation-mediated

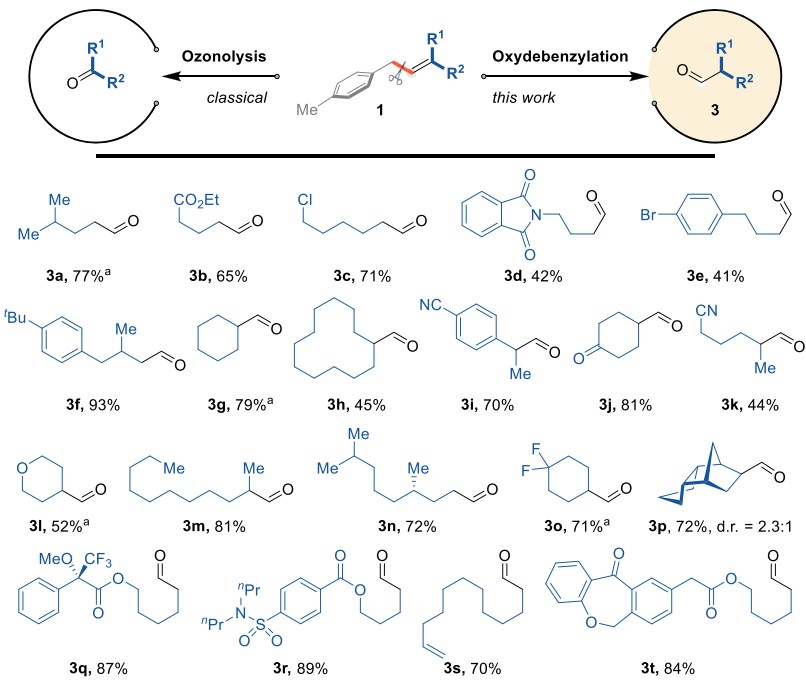

**Fig. 3 | Synthesis of aldehydes by redox-neutral cleavage of allylbenzenes. a** Yields were determined by GC because of the low boiling points of these products.

**A. Light on-off experiment**

**B. Effect of light intensity**

**C. Radical trapping experiment**

**D. Deuterium labeling experiment**

*d*-**4a**, **D**/H (1.68/0.32)
migration & HAT twice

**E. Involvment of reaction intermediates**

olefin migration

std. cond. 84%

std. cond. 98%

std. cond. 93%

**F. Substrates lack the ability to migrate**

sub. **4b**

std. cond.
No cleavage product

sub. **4c**

std. cond.
No cleavage product

**G. A solution to non-migratory substrates**

non-migratory
No reaction

**trisubstituted olefin redox active**

std. cond.

*homoallylic arenes*

non-migratory
No reaction

**trisubstituted olefin redox active**

std. cond.

complicating benzylic protons

**4c**, 75%

**4d**, 60%

**4e**, 55%

**4f**, 64%

**4g**, 80%

**4h**, 52%

**Fig. 4 | Mechanistic investigations. A** A light-on-off experiment ruled out the involvement of radical chain mechanisms **B** The effect of light intensity to the reaction rate. **C** Trapping of the resulting benzyl radical using benzalmalononitrile. **D** A deuterium labeling experiment to support the process of migration. **E** Control exprimrents to confirm the generation of key intermediates. **F** Substrates lack the ability to migrate failed to produce the product. **G** A complementary solution to cleave the C(sp$^2$)-C(sp$^3$) bonds.

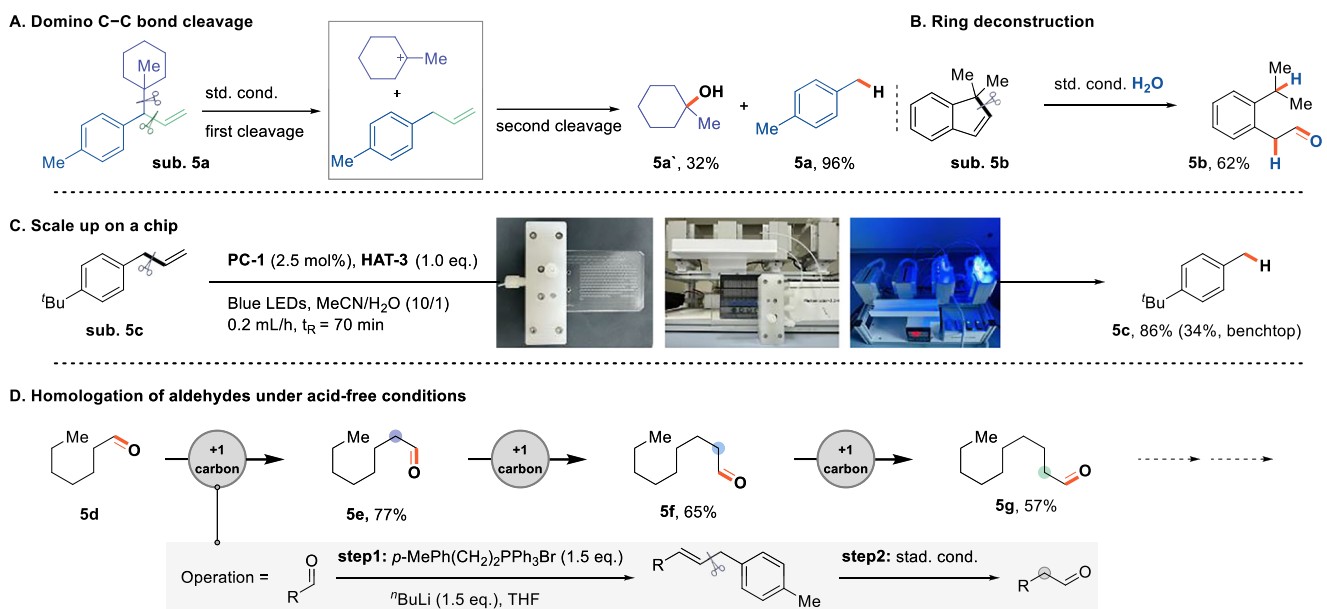

**Fig. 5 | Synthetic utilities. A** Domino C-C bond cleavage to deliver doubly cleaved products in a single reaction. **B** Ring deconstruction of a endocyclic olefin. **C** Standard reation on a chip. **D** Homologation of aldehydes without the addition of acid.

carbon-carbon bond cleavage exhibits tunable selectivity, which can be modulated by adjusting the reaction conditions and substrate substitution. This adaptability provides a strategy for programmable domino bond cleavage reactions, offering a versatile approach to selective bond scission. When a large alkyl group is present at the benzylic position, CCCET predominates over alkene migration, facilitating efficient cleavage of the homobenzylic $C(sp^3)$ $-C(sp^3)$ bond. Following this initial cleavage, the allylbenzene fragment undergoes migration and subsequent $C(sp^2)-C(sp^3)$ bond scission, delivering a doubly cleaved product in a single reaction (Fig. 5A). Further control experiments using two potential intermediates revealed that $C(sp^3)-C(sp^3)$ bond cleavage does not proceed without the corresponding $C(sp^2)-C(sp^3)$ bond, confirming that $C(sp^3)-C(sp^3)$ cleavage precedes $C(sp^2)-C(sp^3)$ cleavage, consistent with prior mechanistic studies (Supplementary Fig. 9)[31].

Our methodology extends its applicability to endocyclic olefins, providing a versatile approach for ring deconstruction and scaffold reorganization (Fig. 5B). To further demonstrate its synthetic appeal, we employed continuous-flow microfluidic chips, which significantly accelerated the reaction rate and improved yield (Fig. 5C, 86% yield *vs.* 34% obtained using a benchtop setup within 70 minutes). An iterative homologation approach was achieved using a series of Wittig/dealkenylation steps (Fig. 5D). This methodology circumvents the necessity of employing harsh acidic or oxidative conditions typically associated with traditional homologation procedures. Using $^{18}$O-labeled water offers an efficient route for synthesizing and homologating $^{18}$O-labeled aldehydes, demonstrating the versatility and applicability of this approach (Supplementary Fig. 4).

## Discussion

This study introduces a mild protocol for the hydrodealkenylation of arylalkenes, enabling selective cleavage of nonpolarized $C(sp^2)-C(sp^3)$ bonds under photoredox conditions. We propose and validate an SEO-mediated alkene migration mechanism that facilitates this process. The method effectively removes a vinyl group from hydrocarbon substrates, demonstrating selectivity among various carbon-carbon bonds of similar strength, with both cleaved fragments isolated in good yields. This redox-neutral approach represents a significant advancement over traditional oxidative protocols by enabling the

cleavage of $C(sp^2)-C(sp^3)$ bonds in mono-substituted alkenes (devinylation). Our reactions demonstrate significant adjustability and synthetic adaptability, featuring electron-density-controlled selectivity, ring deconstruction, domino cleavage of two inert carbon-carbon σ-bonds, and iterative homologation of carbon chains. We believe these findings will have a wide range of applications and stimulate further advancements in the challenging area of inert bond cleavage reactions.

## Methods

### General method for standard condition

PC-1 (0.005 mmol, 2.9 mg) was weighed in an oven-dried 8 mL vial equipped with a magnetic stirring bar. $H_2O$ (0.1 mL) and MeCN (1.0 mL) were added, followed by HAT-3 (0.2 mmol) and the substrate of alkene (0.2 mmol). The reaction vessel was degassed, back-filled with argon, and placed between two kessil lights (40 W*2; refer to Supplementary Fig. 1 in the Supplementary Information for pictures of the reaction setup). An oil bath was used to heat at 40 °C to ensure stable temperature control. TLC monitored the progress of the reaction. Upon completion, the reaction mixture was concentrated and purified by silica gel flash column chromatography.

## Data availability

Experimental procedures and characterizations of new compounds are included in the Supplementary Methods. For NMR and HPLC spectra of structurally novel compounds, see Supplementary Figs. All other data from the authors are available upon request.

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

## Acknowledgements

This work was financially supported by the Guangdong Basic and Applied Basic Research Foundation (2025B1515020070, J.C.), Hong Kong RGC (16305523, Y.H.), the National Natural Science Foundation of China (21825101, Y.H.) and the China Postdoctoral Science Foundation (2024M762152, K.L.).

## Author contributions

J.C. and Y.H. directed the project. K.L., C.G. and Z.C. conducted the experiments. All authors contributed to analyzing the experimental results, as well as to the writing of this manuscript.

## Competing interests

The authors declare no competing interests.
