## [Transparent Peer Review file · Nature Communications]

Redox-neutral photocatalytic hydrodealkenylation of aryl olefins

Corresponding Author: Professor Jean Chen

Version 0:

Reviewer comments:

Reviewer #1

(Remarks to the Author)

Liao et al. reported the hydrodealkenylation of various aryl olefins using a photocatalyst. Recently, carbon-carbon bond cleavage reactions have garnered significant attention in the field of reaction methodology development. The removal of vinyl groups is particularly challenging under mild reaction conditions. In this study, the authors have elegantly demonstrated an interesting transformation that enables C–C bond cleavage.

Table 1 presents the optimization of reaction conditions. The combination of acridinium photocatalyst and HAT reagent 3 resulted in the highest yield. The substrate scope is summarized in Table 2, which is well-organized. More interestingly, the authors focused on the aldehyde fragment to highlight its synthetic applicability. Various mechanistic studies shown in Figure 2 support the proposed reaction mechanism.

Overall, this work meets the criteria for publication in Nature Communications after minor revisions.

Comments:

1. Figures are difficult to interpret

The figures are highly challenging to understand. In particular, in Figure 1C, the structures are labeled as R1–R3, making it difficult to grasp the novelty of the work. Furthermore, in Figure 1D, the transformation from the alkenyl radical cation to the final product is not clearly depicted, making the mechanistic pathway hard to follow. Additionally, Figures 3A and 3D lack sufficient descriptions of the specific reactions being performed, which hinders comprehension. Given the intriguing nature of the study, the authors should improve the presentation of figures to convey their findings better.

2. Numerous errors in the Supporting Information (SI)

The SI contains excessive errors, to the extent that I even considered rejecting the manuscript solely on this basis. For example, in the first equation of 2. Preparation of substrates, “N3Et” is incorrectly written instead of the correct “NEt3.” This oversight raises concerns about whether the SI was properly reviewed by the authors. Similarly, in General Procedure F on the following page, the equivalent amount of PPh3 is missing.

In the characterization of compound 3c, the notation “1.59 (p, J = 7.4 Hz)” appears, but what does “p” refer to? Additionally, in 4.4 Plausible Mechanism, the formation of 2-methylpropanal from Int IV contains structural errors. Furthermore, in the section on trisubstituted alkenes, many structures are missing, making the content nearly incomprehensible.

The authors should carefully review and correct these issues before publication.

Reviewer #2

(Remarks to the Author)

Reviewed is an intriguing paper from Chen, Huang, and coworkers on the development and study of a photocatalytic alkene migration/cleavage reaction allowing for styrenes, allylbenzenes, and trisubstituted alkenes to be cleaved, resulting in coproduction of alkane (typically toluyl) and aldehyde. The proposed and quite intriguing mechanism of the reaction leverages the extreme oxidation potential of acridinium photocatalysis to access radical cation intermediates that allow allylbenzenes to be isomerized to styrenes in the presence of an electron deficient HAT donor. Following this, a hydration/oxy radical beta-cleavage cascade terminated by a HAT reaction is proposed to generate the alkane and

aldehyde. This proposal is well-supported by the mechanistic studies in figure 2, and I applaud the authors for performing such extensive studies to test the reasonableness of intermediates and requirements for alkene migration. With respect to synthetic applications, the reaction is relatively specialized (requiring styrene, allylbenzene, or trisubstituted alkene), but produces high yield generally for successful substrates.

Taken together, this is a challenging, albeit somewhat specialized, reaction that proceeds through an interesting and unique mechanism. I believe the quality and novelty of the mechanistic study and uniqueness of the disconnection elevate this study to the level expected of Nature Communications and support publication of a suitably edited manuscript.

Reviewer #3

(Remarks to the Author)

The manuscript presents a novel redox-neutral approach for hydrodealkenylation, enabling selective cleavage of C(sp²)-C(sp³) σ -bonds in aryl olefins under photoredox conditions. The study demonstrates synthetic versatility, selectivity, and mechanistic insights into the reaction pathway. Additionally, the system appears to be highly sensitive to electronic effects, with the substrate scope mainly limited to electron-rich aryl olefins. Further elaboration is necessary to clarify these aspects.

More comments:

1. The Stern-Volmer quenching experiments and time-course reaction monitoring are crucial for understanding the reaction kinetics. These experiments should be included to clarify the interaction between the photocatalyst and the substrates/intermediates.
2. The proposed mechanism suggests a key migratory intermediate before bond cleavage. The authors should attempt to capture this intermediate using radical trapping, in-situ spectroscopy, or other suitable methods. This would provide stronger support for the proposed reaction pathway.
3. Introducing 1,4-diene-type substrates could be insightful, and the corresponding competition experiments would help establish the selectivity and limitations of the method.
4. In the domino cleavage mechanism, there is no direct evidence proving that the C(sp³)-C(sp³) bond cleavage occurs first, followed by C(sp²)-C(sp³) bond cleavage. Additional experimental validation is needed to confirm this reaction sequence.
5. Inconsistencies in figure and table titles should be carefully reviewed and corrected to maintain uniform formatting.
6. A careful review of the references is required to ensure consistency in formatting and citation style.

Version 1:

Reviewer comments:

Reviewer #1

(Remarks to the Author)

I have read the manuscript as Reviewer 1. The authors have adequately addressed my comments. I believe the manuscript is now suitable for publication.

Reviewer #3

(Remarks to the Author)

I am happy with their corrections and recommend to accept this article

We would like to sincerely thank the editors, as well as the reviewers, for the efficient work and instructive suggestions on "*Redox-neutral photocatalytic hydrodealkenylation of aryl olefins*" (Manuscript ID: NCOMMS-25-07850-T). We have carefully revised this manuscript according to the reviewers' recommendations. The detailed changes and arguments are highlighted in blue.

Reviewers' Comments to Author (Responses are highlighted in blue):

Reviewer #1:

Liao et al. reported the dehydrodealkenylation of various aryl olefins using a photocatalyst. Recently, carbon-carbon bond cleavage reactions have garnered significant attention in the field of reaction methodology development. The removal of vinyl groups is particularly challenging under mild reaction conditions. In this study, the authors have elegantly demonstrated an interesting transformation that enables C–C bond cleavage.

Table 1 presents the optimization of reaction conditions. The combination of acridinium photocatalyst and HAT reagent 3 resulted in the highest yield. The substrate scope is summarized in Table 2, which is well-organized. More interestingly, the authors focused on the aldehyde fragment to highlight its synthetic applicability. Various mechanistic studies shown in Figure 2 support the proposed reaction mechanism.

Overall, this work meets the criteria for publication in Nature Communications after minor revisions.

Our response: We would like to thank Reviewer 1 for his/her recognition of the overall research work and the evaluation of innovation.

Comments:

1. Figures are difficult to interpret

The figures are highly challenging to understand. In particular, in Figure 1C, the structures are labeled as R1–R3, making it difficult to grasp the novelty of the work. Furthermore, in Figure 1D, the transformation from the alkenyl radical cation to the final product is not clearly depicted, making the mechanistic pathway hard to follow. Additionally, Figures 3A and 3D lack sufficient descriptions of the specific reactions being performed, which hinders comprehension. Given the intriguing nature of the study, the authors should improve the presentation of figures to convey their findings better.

Our response: We sincerely appreciate the insightful feedback provided by Reviewer 1, which has significantly strengthened the clarity and impact of our manuscript. Below, we address the reviewer's concerns in detail:

We have meticulously restructured the overall framework of Figure 1 to eliminate redundant information while emphasizing the innovative aspects of our methodology. The

revised figure now prioritizes clarity and conciseness, ensuring readers can readily grasp the core advancements of this work. Figure 1D has been revised to include three critical intermediates identified through mechanistic control experiments and references: benzyl radical, olefin, and alcohol intermediate. This adjustment provides direct experimental evidence to elucidate the catalytic mechanism, aligning with the reviewer's request for enhanced mechanistic transparency.

To improve the clarity of the double cleavage strategy in substrates with multiple C–C bonds, we have reorganized Figure 3A to emphasize the bond activation sequence visually. This revision underscores the selectivity and efficiency of our approach. Additionally, we have explicitly introduced detailed schematics to outline the operational logic of the aldehyde homologation process (Figure 3D). These schematics bridge the conceptual framework with experimental execution, ensuring readers can readily follow the rationale behind our design.

Modification in the revised manuscript:

Figure 1. Cleavage of unactivated carbon-carbon bonds.

Figure 3. Synthetic utilities.

2. Numerous errors in the Supporting Information (SI)

The SI contains excessive errors, to the extent that I even considered rejecting the manuscript solely on this basis. For example, in the first equation of 2. Preparation of substrates, "N3Et" is incorrectly written instead of the correct "NEt3." This oversight raises concerns about whether the SI was properly reviewed by the authors. Similarly, in General Procedure F on the following page, the equivalent amount of PPh₃ is missing. In the characterization of compound 3c, the notation "1.59 (p, J = 7.4 Hz)" appears, but what does "p" refer to? Additionally, in 4.4 Plausible Mechanism, the formation of 2-methylpropanal from Int IV contains structural errors. Furthermore, in the section on trisubstituted alkenes, many structures are missing, making the content nearly incomprehensible.

Our response: We sincerely thank Reviewer 1 for his/her rigorous evaluation and constructive feedback, which have significantly strengthened the scientific quality of our work. We sincerely regret any oversights in the initial submission and have undertaken a comprehensive revision of the Supplementary Information (SI) to address errors, improve clarity, and enhance reproducibility. All experimental protocols, datasets, and analytical methods have been meticulously cross-checked and standardized to ensure accuracy.

Furthermore, we have expanded mechanistic investigations with additional control experiments and provided robust evidence for the proposed reaction pathway documented in the revised SI to improve transparency. The revisions collectively reinforce the scientific rigor of the study, ensuring readers can fully assess the methodology and reproduce the findings.

Reviewer #2:

Reviewed is an intriguing paper from Chen, Huang, and coworkers on the development and study of a photocatalytic alkene migration/cleavage reaction allowing for styrenes, allylbenzenes, and trisubstituted alkenes to be cleaved, resulting in coproduction of alkane (typically toluyl) and aldehyde. The proposed and quite intriguing mechanism of the reaction leverages the extreme oxidation potential of acridinium photocatalysis to access radical cation intermediates that allow allylbenzenes to be isomerized to styrenes in the presence of an electron deficient HAT donor. Following this, a hydration/oxyl radical beta-cleavage cascade terminated by a HAT reaction is proposed to generate the alkane and aldehyde. This proposal is well-supported by the mechanistic studies in figure 2, and I applaud the authors for performing such extensive studies to test the reasonableness of intermediates and requirements for alkene migration. With respect to synthetic applications, the reaction is relatively specialized (requiring styrene, allylbenzene, or trisubstituted alkene), but produces high yield generally for successful substrates.

Taken together, this is a challenging, albeit somewhat specialized, reaction that proceeds

through an interesting and unique mechanism. I believe the quality and novelty of the mechanistic study and uniqueness of the disconnection elevate this study to the level expected of Nature Communications and support publication of a suitably edited manuscript.

Our response: We sincerely thank Reviewer 2 for his/her thoughtful evaluation and recognition of our work. The selective cleavage and functionalization of C_{sp2}–C_{sp3} bonds represent a critical advancement in our broader goal of establishing a systematic framework for inert bond activation.

The current substrate limitation, as noted, arises from the requirement for a single-electron oxidation step to initiate the reaction. This mechanistic constraint necessitates substrates with stabilizing aromatic systems (e.g., benzene rings) or electronically enriched alkenes. We fully acknowledge this challenge and agree with Reviewer 2's insight that extending the methodology to non-aromatic or less-activated alkenes—while significantly more demanding—would substantially broaden its applicability. Though beyond the current scope, such systems represent a compelling and logical direction for future investigation. We appreciate Reviewer 2's emphasis on these fundamental challenges aligning with our long-term research vision. Their suggestions will inform subsequent efforts to refine the mechanistic scope and explore unconventional substrate classes.

Reviewer #3:

The manuscript presents a novel redox-neutral approach for hydrodealkenylation, enabling selective cleavage of C(sp²)–C(sp³) σ-bonds in aryl olefins under photoredox conditions. The study demonstrates synthetic versatility, selectivity, and mechanistic insights into the reaction pathway. Additionally, the system appears to be highly sensitive to electronic effects, with the substrate scope mainly limited to electron-rich aryl olefins. Further elaboration is necessary to clarify these aspects.

Our response: We sincerely thank Reviewer 3 for his/her insightful questions and constructive feedback. Below, we clarify the mechanistic rationale underlying the substrate limitations observed in our study:

The successful cleavage of nonsubstituted and *para*-bromide-substituted alkenes (products **4d**, 60%; **4e**, 55%) under standard conditions highlights the role of electronic effects in our system. While these substrates represent the most electron-deficient cases tested, their reactivity is enabled by the trisubstituted alkene moiety's single-electron oxidation (SEO), which initiates the process.

For monosubstituted olefins, reactivity is contingent on the oxidation of the aromatic ring (mediated by the acridinium photocatalyst; *Chem. Sci.* **2017**, *8*, 4654-4659). Electron-deficient aromatic systems resist this oxidation step, preventing subsequent migration and

hydration. In trisubstituted alkenes, SEO of the C=C bond bypasses this limitation, allowing hydration to proceed. However, the final proton-coupled electron transfer (PCET)-induced cleavage (*Cell Rep. Phys. Sci.* **2022**, *3*, 100763) still requires aromatic ring oxidation, rendering electron-deficient alkenes inactive.

We rigorously tested several electron-deficient substrates under standard conditions, but none yielded the desired products, consistent with the mechanistic constraints outlined above. These findings underscore the nuanced interplay between substrate electronics and catalytic steps, guiding future efforts to expand substrate scope.

More comments:

1. The Stern-Volmer quenching experiments and time-course reaction monitoring are crucial for understanding the reaction kinetics. These experiments should be included to clarify the interaction between the photocatalyst and the substrates/intermediates.

Our response: We would like to thank reviewer 3 for his/her professional suggestions. The Stern-Volmer quenching experiments have been performed. The result confirmed that the alkene was the primary quencher of the excited photocatalyst.

Supplementary Figure 6. Stern-Volmer quenching experiment

Time-course reaction monitoring experiments have been done, and the key intermediate of alcohol (**1a-OH**) can be detected in a low yield during the reaction. The migratory styrene (**1a'**) can also be detected. However, the concentration remained low during the whole reaction process. These results have been appended to the revised manuscript and SI.

Supplementary Figure 5. Kinetic study

2. The proposed mechanism suggests a key migratory intermediate before bond cleavage. The authors should attempt to capture this intermediate using radical trapping, in-situ spectroscopy, or other suitable methods. This would provide stronger support for the proposed reaction pathway.

Our response: We sincerely thank Reviewer 3 for his/her valuable suggestions. TEMPO and benzalmalononitrile were employed as trapping agents under standard conditions to probe the migratory radical intermediate. While these experiments yielded complex mixtures without identifiable trapping products, the further mechanistic investigation by excluding water from the reaction enabled the detection of the migratory intermediate via GCMS analysis ($\approx 5\%$ yield). This observation aligns with kinetic studies confirming the formation of alcohol derivatives originating from migratory styrene, supporting the proposed radical migration pathway.

In addition, step-by-step reaction evaluation, starting from possible intermediate products at different stages, can guide the target cleavage products and prove the mechanism of migration and hydroxylation to a certain extent (Figure 2E).

3. Introducing 1,4-diene-type substrates could be insightful, and the corresponding competition experiments would help establish the selectivity and limitations of the method.

Our response: We would like to thank reviewer 3 for his/her insightful suggestions. Several 1,4-diene-type substrates have been synthesized and evaluated under standard conditions. When the aromatic moiety is retained in the substrate, the 1,4-diene skeleton is well-tolerated, enabling efficient C_{sp^2} - C_{sp^3} cleavage to proceed smoothly and deliver the target product in 78% yield. In contrast, substrates lacking an aromatic moiety fail to

undergo cleavage. Instead, water addition to the double bond dominates, yielding mixtures of alcohol derivatives.

4. In the domino cleavage mechanism, there is no direct evidence proving that the C(sp³)-C(sp³) bond cleavage occurs first, followed by C(sp²)-C(sp³) bond cleavage. Additional experimental validation is needed to confirm this reaction sequence.

Our response: We sincerely thank Reviewer 3 for his/her constructive feedback. Reactions were conducted using two potential intermediates as substrates to validate the proposed mechanism. These experiments revealed that C_{sp3}-C_{sp3} bond cleavage does not proceed without the corresponding C_{sp2}-C_{sp3} bond (**Int-I**). This observation aligns with our finding that 4-allyl toluene undergoes efficient sequential bond cleavage under standard conditions, achieving a 93% yield of the target product. These results confirm that C_{sp3}-C_{sp3} cleavage precedes C_{sp2}-C_{sp3} cleavage, consistent with prior mechanistic studies (*J. Am. Chem. Soc.* **2023**, *145*, 12284-12292). The carbocation-coupled electron transfer (CCET) process requires a substituent at the benzyl position to stabilize intermediates, as unsubstituted analogs fail to react. Now integrated into the revised manuscript and SI, these insights clarify the role of the bond activation sequence and substituent effects in governing reactivity.

5. Inconsistencies in figure and table titles should be carefully reviewed and corrected to maintain uniform formatting.

Our response: We appreciate the reviewer's reminder and apologize for the inconsistencies in the figure and table titles. The format has been standardized in the revised manuscript and SI.

6. A careful review of the references is required to ensure consistency in formatting and citation style.

Our response: We appreciate the reviewer's reminder and apologize for the inconsistencies. The reference formatting has been standardized in the revised manuscript and SI.